# Performance Analysis of Linearly Arranged Concentric Circular Antenna Array with Low Sidelobe Level and Beamwidth Using Robust Tapering Technique

**DOI:** 10.3390/mi13111959

**Published:** 2022-11-11

**Authors:** Imteaz Rahaman, Md Ashraful Haque, Narinderjit Singh Sawaran Singh, Md. Shakiul Jafor, Pallab Kumar Sarkar, Md Afzalur Rahman, Mohd Azman Zakariya, Ghulam E. Mustafa Abro, Nayan Sarker

**Affiliations:** 1Electrical Engineering, Ingram School of Engineering, Texas State University, San Marcos, TX 78666, USA; 2Department of Electrical and Electronic Engineering, Universiti Teknologi PETRONAS, Seri Iskandar 32610, Perak, Malaysia; 3Department of Electrical and Electronic Engineering, Daffodil International University, Dhaka 1341, Bangladesh; 4Faculty of Data Science and Information Technology (FDSIT), INTI International University, Per-Siaran Perdana BBN, Putra Nilai, Nilai 71800, Negeri Sembilan, Malaysia; 5Department of Electrical and Electronic Engineering, Rajshahi University of Engineering Technology, Rajshahi 6204, Bangladesh; 6Condition Monitoring Systems Lab, NCRA, Mehran University of Engineering and Technology (MUET), Jamshoro 76020, Pakistan; 7Department of Electronics and Communication Engineering, Khulna University of Engineering & Technology (KUET), Khulna 9203, Bangladesh

**Keywords:** LCCAA, sidelobe level, beamwidth, interference, optimal, robust techniques, FDL, ODL, VDL, tapering techniques–uniform, binomial, Chebyshev, Blackman, Hamming, Hanning, Taylor, triangular, robust tapering technique

## Abstract

In this research, a novel antenna array named Linearly arranged Concentric Circular Antenna Array (LCCAA) is proposed, concerning lower beamwidth, lower sidelobe level, sharp ability to detect false signals, and impressive SINR performance. The performance of the proposed LCCAA beamformer is compared with geometrically identical existing beamformers using the conventional technique where the LCCAA beamformer shows the lowest beamwidth and sidelobe level (SLL) of 12.50° and −15.17 dB with equal elements accordingly. However, the performance is degraded due to look direction error, for which robust techniques, fixed diagonal loading (FDL), optimal diagonal loading (ODL), and variable diagonal loading (VDL), are applied to all the potential arrays to minimize this problem. Furthermore, the LCCAA beamformer is further simulated to reduce the sidelobe applying tapering techniques where the Hamming window shows the best performance having 17.097 dB less sidelobe level compared to the uniform window. The proposed structure is also analyzed under a robust tapered (VDL-Hamming) method which reduces around 69.92 dB and 48.39 dB more sidelobe level compared to conventional and robust techniques. Analyzing all the performances, it is clear that the proposed LCCAA beamformer is superior and provides the best performance with the proposed robust tapered (VDL-Hamming) technique.

## 1. Introduction

In this modern world, technology in the communication industries has enhanced control over time and distance in the last few decades. Thus, it is a mandatory feature of today’s antenna system to be smart. Antenna array beamforming offers high directivity, narrow beamwidth, low side-lobes, point-to-point, and preferred–coverage pattern characteristics. Beamforming multiplies signals from each antenna element with a specified weight and combines signals from array elements of an array antenna system [1]. The phased antenna array has been considered a prominent research topic for its long-desirable applications such as satellite communication systems, radar, underwater communication, sonar, satellite communication, and radio astronomy. The exceptional characteristic of the phased array is its prompt and flexible beam scanning, where the main beam is steered towards a particular direction electronically [2]. Among many more antenna array structures, linear arrays and planar arrays are widely used due to their easy control for 3D beamforming [3].

Although the linear array exhibits a low sidelobe level (SLL) radiation pattern in any given direction [4], the circular antenna array is considered the perfect configuration as its main lobe can steer in all azimuth directions without changing its bandwidth [5]. However, inter-element space unexpectedly results in a mutual coupling effect. To minimize this problem, a unique structure named concentric multi-ring array with sufficient inter-element spacing is introduced. Concentric circular antenna array (CCAA) has the capability of all-azimuth scanning and the beam pattern remains circularly symmetric in this structure, which is invariant for 360° azimuthal coverage circularly symmetric [4]. Moreover, a concentric circular antenna array (CCAA) performs better in SLL reduction than a circular antenna array (CAA) concerning the same number of elements [6]. Similarly, a uniform concentric circular antenna array (UCCAA) is another structure where the array element is arranged at a fixed distance which is normally 0.5$\lambda_{0}$. Usually, $\lambda_{0}$ represents the wavelength of the highest frequency when it comes to the reduction of SLL in high-frequency radiation patterns [7]. Though UCCAA holds simple circuitry and it is easy to implement, the SLL is comparatively higher. So, to obtain better directivity, a large-scale UCCAA is the alternative solution [8].

The cat swarm optimization (CSO) algorithm is applied in the nine-ring time modulated concentric circular antenna array (TMC-CAA), which claims that the increment of sideband frequency causes a decrease of the power radiated by harmonic frequencies and SLL [9]. Moreover, tapering techniques are applied to reduce SLL in the LCCAA beamformer where the Hamming window shows the best result, though the performance under SINR to SNR is completely ignored [10]. Additionally, a brief introduction to the particle swarm optimization (PSO) technique for designing multi-ring concentric circular arrays (CCA) and concentric hexagonal arrays (CHA) is provided [11]. The radius of the ring and the inter-ring spacing of a CCAA system are optimized by applying an adaptive technique to achieve a circularly symmetric pattern [12,13,14]. A hybrid method is proposed to synthesize a concentric ring array (CRA) which is numerically stable and computationally efficient [3]. Moreover, the phase-only beamformer performs well as a direction of arrival (DOA) estimator at low SNR [15]. Apart from this, a spherical hydrophone array improves the SNR for underwater beamforming [16]. Convex optimization and a deterministic approach are proposed as hybrid methods for sparse concentric ring arrays which can optimize both SLL and first null beamwidth (FNBW) [17]. A novel vertically polarized (V-pol) planar folded slot antenna is proposed to incorporate an energy-efficient 5G phased array for user devices are presented in [18]. The performance is analyzed for both circular and concentric circular antenna arrays, applying the DOA estimation technique in the presence of high and low noise environments [19,20]. In this approach, robust strategies are completely disregarded in favor of the best technique, which is used to compare performance. On the basis of a diversely polarized antenna (DPA) array, a method is proposed for the performance analysis of space-time adaptive processing (STAP), where the proposed method offers improved clutter suppression performance and carries quasi-convex form [21]. The decreased performances are examined while taking into account the mismatch between the guided signal and the actual signal. The aforementioned studies [22,23] cover the resilience of the antenna array with regard to SNR performance, however they do not cover the SLL reduction technique or algorithm. The implementation of a smart antenna system and their monitoring method are briefly detailed in Ref. [24], and a unique signal processing framework is addressed where the array time samples are taken into account in the DOA-frequency scheme using a single-stage problem [25]. Additionally, two novel techniques, the unscented transform (UT) approach and the principal component analysis (PCA) method, are briefly addressed [26]. Additionally, compressive antenna arrays are briefly described for estimating narrowband DOA in order to determine the larger aperture and lower hardware complexity [27]. When the interference signal can be detected and attenuated by the FDL, however, the limits of the conventional approach are described in terms of interference detection for a CCAA beamformer [28]. In fact, references [29,30,31,32] provide a brief overview of current robust techniques for various array geometries.

For the antenna array outlined above, there are still certain areas where SLL and beamwidth performances can be enhanced. Additionally, the aforementioned articles lack a comparison scheme of various antenna arrays, which could clarify the fundamental understanding for selecting appropriate antenna arrays. Most notably, the performance has not yet improved from the robust tapering strategy. We attempt to propose an antenna array that, in comparison to other current antenna arrays, has the best SLL, beamwidth, interference cancellation, and enhanced SINR characteristics. A unique robust tapering technique (VDL-Hamming) is suggested in this regard. This study presents a comparison of geometrically comparable antenna arrays. In this paper, a beamformer named Linearly arranged Concentric Circular Antenna Array (LCCAA) is proposed to overcome the aforementioned difficulties.The proposed beamformer is compared to the current beamformer-concentric circular antenna array (CCAA), linear array [2], uniform concentric circular antenna array (UCCAA), centered element concentric circular antenna array (CECCAA), uniform centered element concentric circular antenna array (UCECCAA), and uniform linearly arranged concentric circular antenna array (ULCCAA). The proposed LCCAA beamformer performs better than the existing beamformers in all comparisons. The optimal technique is used to peak the actual signal and reduce interference because the conventional technique is unable to peak the rightmost signal. However, when there is a look direction error, the optimal technique has a larger SLL. This proposed beamformer is further analyzed by applying robust techniques- ODL, FDL, and VDL which reduce beamwidth to a great extent while the SLL performance remains poor. In this regard, tapering techniques are applied to reduce the sidelobe level where it is observed that the Hamming tapering technique shows the optimum capability for the reduction of SLL. In this research work, a robust tapered (VDL-Hamming) technique is proposed by the combination of robust and tapering techniques for the proposed structure. It is analyzed under a robust, conventional, and robust tapering technique. The proposed processor has the following desirable properties.

The proposed LCCAA beamformer can scan 360°.The proposed LCCAA beamformer is typically a robust beamformer against look direction disparity.It can cancel directional interferences.It has lower beamwidth of 12.482° for VDL robust technique with 102 elements.The proposed beamformer is capable of maximizing the SINR.Robust VDL technique enhances the output power of 29.02 dB, 18.66 dB, 0.77 dB compared to conventional. FDL and ODL robust techniques.Different tapering techniques are applied in this proposed beamformer to reduce the SLL. It is improved by using the Hamming window which reduces, by −35.279 dB, the sidelobe level for three rings.Finally, the robust tapered (VDL-Hamming) technique is proposed in this research which reduces around 69.92 dB and 48.39 dB more SLL compared to conventional and robust techniques.

## 2. System Model

In an antenna array, peaking false signals is the most frequent problem as the antenna array has to scan 360°. The circular antenna array is the strongest solution in this regard. Figure 1 shows a complete communication system using a centered element concentric circular antenna array (CECCAA) beamformer of 34 elements.

In this block diagram, an adjustable weighting vector is used whose task is to peak up the rightmost signal. This weighting vector is applied to create null in the unexpected signal direction which helps to eradicate unwanted signals. After receiving the signal from each antenna element, this weighting vector has to multiply with these signals. The weight is always calculated concerning the desired direction. A complete arrangement of the antenna element in a circular antenna array structure with three rings along with one central element is pictured in Figure 2.

The innermost ring consists of five elements, the middle ring contains 11 elements, and the outermost ring has 17 elements. The antenna elements are uniformly distributed with a radius of 0.5λ. As shown in Figure 2, the center of the CECCAA structure is at the origin where L denotes the antenna element. The radius is considered (as rm is the radius of the mth ring) and all the elements are equally spaced. The ring number is symbolized by ‘m’ and the letter ‘M’ is used to denote the maximum ring number.
(1)τ=((x1−x2)cosα+(y1−y2)sinα)/d

### 2.1. Antenna Arrays

Antenna arrays are the fundamental element that mainly increases the output power, gain, and directivity. In this research, a few circularly structured antenna arrays are considered for their similar geometric structure, including the linear antenna array. In Figure 3a, the linear antenna array is shown, where 21 elements are used. The spacing between the two adjacent elements is 0.5λ.

The major flaw in this technology is its linear antenna array, which frequently peaks at erroneous signals. As a result, an obvious necessity arises, for scanning any object 0° to 360° peaking the rightmost signal. In the linear structure, the length is increased with the increase of the antenna element. For the LCCAA beamformer, the length of the central element of each ring will be equal to the length of the linear array. In Figure 3b–f shows the antenna array using a circular array structure with changing the element position.

In Figure 3b,c, the CCAA, and UCCAA structures are enunciated where the antenna element is 33. In this figure three array ring is considered where the inner ring contains five elements, the middle ring has 11 elements, and the outermost ring has 17 elements. In the same way, the UCCAA structure has 33 elements but each ring contains 11 elements equally. The distance between the two elements is 0.5λ. The CECCAA and CCAA structure is the same but have one element in the center for the CECCAA beamformer which is shown in Figure 3d,e.

Three CCAA structures are placed linearly in the ULCCAA structure. Even though the centered element only raises one element, it significantly affects beamwidth and sidelobe level. Thus, the identical antenna element layout is taken into consideration with just one element in the middle.

### 2.2. Proposed Model

Figure 4 provides a brief overview of the LCCAA beamformer’s system model. Three concentric circular antenna arrays with the core elements are linearly aligned in this beamformer. Additionally, each ring’s central element forms a linear array. This novel beamformer is known as the linearly arranged concentric circular antenna array as a result (LCCAA). A total of 34 elements make up each CCAA beamformer with one center element. Figure 4 shows the inner, middle, and outer rings as well as the central element.

Each CCAA beamformer consists of three rings where the inner ring contains five elements, the middle ring has 11 elements, and the outermost ring contains 17 elements and, most importantly, all rings share one central element. In this proposed beamformer, the middle element of each ring creates a linear structure of 21 elements. The total number of elements considered is 102 in this proposed beamformer. In this proposed model, the distance between two adjacent elements is 0.5λ. Furthermore, the distance between the two rings is also considered about 0.5λ.

## 3. Methodology

The whole methodology of this research is shown in this section. Signal inducing is the first step to be started from. The induced signal is then used to establish a steering vector. After this, the correlation matrix should be formed. Then the weighting vectors are applied to the signal. After multiplying the weighting vector with the actual signal direction, we finally obtain the output power. $m_{k}e^{j2\pi ft}$, denotes the signal, induced on any array elements for $K_{th}$ source at any time. Moreover, $\eta \;(\phi_{k},\theta_{k}$) will be the pre-requisite time for the $l_{th}$ element for the signal from $K_{th}$ sources when it induces at an angle ($\varphi_{k},\theta_{k}$). The necessary time will be $m_{k}\left(t\right)e^{j2\pi f(t-\eta \left(\varphi_{k},\theta_{k}\right))}$ for any individual element, and denote complex modulation frequency and carrier frequency accordingly. The combination of background noise and the incidence of signal from the $l_{th}$ element with *N* directional sources are prominent steps in this beamforming technique.

The combination of background noise and the incidence of signal from the $l_{th}$ element with *N* directional sources are shown in Equation (Equation 2).
(2)$$x_{l}\left(t\right)=\sum_{k=1}^{N}m_{k}\left(t\right)e^{j2\pi f(t-\eta \left(\phi_{k},\theta_{k}\right))}+n_{l}\left(t\right)$$
$n_{l}\left(t\right)$ is applied for the noise component at the $l_{th}$ element and the uncorrelated noise with directional sources is calculated by $E[m_{k}\left(t\right)n_{l}\left(t\right)]=0.$ The presence of noise and interference is the key factor to create a massive difference in the array correlation matrix. Equation (Equation 3) represents the array correlation matrix [31].
(3)$$R=p_{s}S_{0}S_{0}^{H}+p_{I}S_{I}S_{I}^{H}+\sigma_{n}^{2}I$$
$p_{s},p_{I},\sigma^{2}$ are used for signal power, interference signal, and random noise respectively. $S_{0}$ is the steering vector in the look direction and *S* is symbolized for interferences. The weights of the array are estimated by Equation (Equation 4) [32].
(4)$$W_{c}=\frac{1}{L}S_{0}$$

The steering vector in the look direction is indicated by $S_{0}$. The response of the processor in the direction (ϕ,θ) with steering vector S (ϕ,θ) is calculated by y(ϕ,θ) Equation (Equation 5).
(5)$$y\left(\phi ,\theta \right)=W_{c}^{H}S_{0}$$

The methodology is divided into two parts: beamforming technique and tapering technique.

### 3.1. Beamforming Techniques

Using a weighting vector, the beamforming technique expands the field of antenna arrays. Robust approaches use the weighting vector to apply null in the direction of unwanted signals. It is named after robust methods that can identify erroneous signals and cancel out the noise as a result. The three most common techniques is-diagonal loading are Fixed (FDL), Optimal (ODL), and Variable (VDL). The theory of all the beamforming techniques employed in this study is briefly covered in Refs. [2,12,31,33].

### 3.2. Tapering Techniques

In many applications such as radar, sonar low side lobe level (SLL) is required for better performance. Tapering Techniques are basically used for reducing sidelobe level. The applied tapering technique is discussed in this section. In Refs. [12,31], the theory underlying different tapering techniques are briefly outlined.

## 4. Performance Analysis

The whole performance is analyzed in terms of 2D directivity, and the performance by applying conventional techniques, robust techniques, tapering techniques, and robust tapering techniques are also observed.

### 4.1. Directivity

The 2D radiation pattern of the Linear and LCCAA beamformer is shown in Figure 5. The signal direction and steering direction is considered at 50° for all the structure. The signal frequency is taken as 300,000 Hz. All the structures have an equal number of elements: 102.

From Figure 5, we find the directivity of 20.1 dB for the linear beamformer where the LCCAA beamformer shows 17.37 dB directivity. Though the directivity of the linear beamformer is better than the LCCAA beamformer, the linear array shows gratings at 50° and 130°. That means a linear array can peak the false signal. However, there is no such mirror signal in the LCCAA beamformer. Therefore, whereas the linear array is unable to peak the rightmost signal, the LCCAA beamformer can.

### 4.2. Performance Analysis Applying Conventional Technique

A comparison between the LCCAA beamformer and other beam-formers are shown in Figure 6 were linear, UCCAA, CCAA, LCCAA, and ULCCAA beamformers have 102 elements but CECCAA and UCECCAA structure possess 103 elements for having one element in the center. The steering angle is considered at 50° where the actual signal direction is at 52°. The interference angle is taken at 30°. The conventional technique is applied in Figure 6, to analyze the performance of the corresponding beamformer.

In this figure, the linear beamformer has a beamwidth of 4.64° at the output power of −20 dB and sidelobe level of −24.08 dB. On the contrary, the LCCAA beamformer has a beamwidth of 12.4° at −20 dB output power and the sidelobe level is −15.85 dB. However, having 102 elements, the linear beamformer is practically not fit to be used. Moreover, gratings are seen in the linear beamformer which causes the peaking of a false signal. So, the LCCAA beamformer is a better solution in this regard. The comparison of all structures having an equal element for azimuth plane applying the conventional technique is shown in Figure 7.

In Figure 7, the number of elements is the same as in Figure 6 where the comparison takes place in the azimuth plane. In this figure, the beamwidth is larger than the normal plane for all beamformers. The beamwidth is 0.06 dB large in the orthogonal azimuthal plane than a normal plane for the LCCAA beamformer.

Similarly, the beamwidth and the sidelobe level are shown in Table 1 for all beamformers. The beamwidth and SLL are compared for both beamformers with equal elements on the normal plane and orthogonal azimuthal plane.

From Table 1, it is observed that the linear beamformer shows a better result but, practically, the structure is unusable. Alternatively, the LCCAA beamformer shows better performance having a lower beamwidth of 12.50° and 17.88° for identical element spacing and equal element beamforming structure. In Figure 8, the power pattern of the LCCAA beamformer is discussed with the conventional technique and optimal technique.

In this comparison, the steering angle is considered at 50° while the signal direction is at 52° and the interference angle is at 30°. An optimal beamformer exhibits the capability to detect and attenuate interferences as there is a sharp attenuation towards the direction of interference at 30° and signal direction at 52°. Conventional beamformers, in contrast, demonstrate no such kind of ability to detect and attenuate interference. Nevertheless, the problem with the optimal beamformer arises in the presence of a mismatch between steering direction and signal direction. From Figure 8, we can observe that normalized output power in the signal direction is approximately −85 dB, which is lower than the expected normalized power in the signal direction. To resolve the problem of mismatch, different robust techniques can be employed.

### 4.3. Performance Analysis Applying Robust Techniques

Figure 9a shows the comparison of all beamformers for the same element using the ODL technique. Due to the gigantic size, the linear beamformer is excluded from this comparison. The proposed beamforming model LCCAA shows the least beamwidth and least sidelobe level of 12.34° and −13.164 dB respectively. Figure 9b portraits a comparison between the proposed beamformer and the existing beamformer using the VDL technique. After applying the VDL technique, the proposed LCCAA beamformer again shows the least beamwidth. From this figure, it is observed that the sidelobe level is comparatively lower in the VDL technique than in the ODL technique.

Table 2 is formed to describe the beamwidth and sidelobe level for ODL and VDL robust techniques.

From the above table, it is observed that LCCAA provides better performance for both beamwidth and sidelobe levels. Considering all, we can claim that the LCCAA beamformer is superior than other beamformers. When compared to an LCCAA beamformer, the ULCCAA offers superior sidelobe level performance. In terms of structure and performance, the ULCCAA beamformer is close to the LCCAA beamformer. ULCCAA beamformer can therefore works well as an antenna array. For a conclusion, more investigation is required.

Figure 10a,b depicts the variation of the output SINR concerning dis-parity angle for ODL and VDL robust technique, respectively.

From Figure 10a,b, it is observed that the performance of all beamformers is improved using the robust ODL and VDL techniques. The performance degradation problem of the LCCAA beamformer is solved more efficiently using the robust VDL technique than the ODL technique. The output power of the LCCAA beamformer is −0.951 dB in 3° disparity using the ODL technique while the VDL technique has −0.183 dB output power which is 0.768 dB greater than ODL robust technique. In comparison with all existing beamformers, LCCAA provides a better result.

In Figure 11a, the LCCAA beamformer shows a handy performance. The output power of LCCAA beamformers with 1°, 2° and 3° disparity is −0.373 dB, −0.69 dB, and −0.951 dB, respectively. Figure 11b proves the superiority of the proposed LCCAA beamformer applying the VDL technique. So, we can claim that LCCAA performs better with a robust VDL technique. The performances of all existing beamformers are shown in Table 3 using robust ODL and robust VDL technique. It is observed from the table that the performance of an LCCAA beamformer can be improved by using robust ODL and VDL techniques. The table shows that the LCCAA beamformer has the lowest power of −0.023 dB, −0.08 dB, −0.183 dB in 1°, 2°, and 3° disparity angles, using the robust VDL technique, respectively. The performance is categorized from top to bottom as the less efficient to the most as ULCCAA, CECCAA, CCAA, UCCAA, UCECCAA, and LCCAA from top to bottom.

Output SINR with noise power variation is shown in Figure 12a. This figure shows that the output SINR of the robust LCCAA beamformer based on VDL is higher than that of other beamformers currently in use, even when noise power and input SNR are changing. Figure 12b compares the output power of an optimum, FDL, ODL, and VDL technique-based LCCAA beamformer with varying disparity angles.

From Figure 12b, it is also clear that as the disparity angle between the actual signal direction and steering direction increases, the performance of the optimal-based LCCAA beamformer declines. This figure shows that, in the presence of look direction error, VDL robust strategies perform better than other robust techniques.

Table 4 describes the Output Power comparison of Optimal, FDL, ODL, and VDL technique at different disparity angles.

From Table 4, it is clear that the VDL technique shows the lowest deviation of output at each disparity. Without disparity, all robust techniques show 0 dB disparity while for 3° disparity, the optimal, FDL, ODL, and VDL technique show −29.21 dB, −18.85 dB, −0.96 dB and −0.19 dB, respectively, where the VDL technique shows superior performance.

### 4.4. Performance Analysis Applying Tapering Techniques

Table 5 and Table 6 show the sidelobe level and beamwidth comparison of tapered beamformers having different windows with the variation of the number of rings.

The comparison of the sidelobe level in Table 5 is measured with respect to the different numbers of rings. Using the tapering approach, the sidelobe level is reduced in a very desirable manner. The sidelobe level is reduced by all of the windows. However, our goal is to determine which method of sidelobe reduction performs the best. Table 5 makes it very evident that the Hamming window is the most effective at lowering the SLL. The Hamming window displays the lowest SLL for three rings as well as a thorough performance with ring quantity variations. The numeric value of the SLL shows an inconsistent flow with the increasing of ring number. On the other hand, Table 6 portraits the beamwidth reduction scheme for all the tapering techniques. For the Hamming window, the largest beamwidth is 62.09° at the output power of 20 dB for using one ring. However, with the variation of ring number, the beamwidth is decreased due to the increment of the array element. From Table 6, it is clear that, with the increase in the inner ring element, the beamwidth of the beamformer is decreased where all the beamwidth is taken at the 20 dB output power. Figure 13 shows a competitive study of output power with a conventional, binomial, and Hamming window where sidelobe reduction ability is vital.

Figure 13 reveals that the sidelobe level is 17.097 dB less in the Hamming window technique as compared to the conventional window and binomial tapering shows −11.441 dB SLL.

### 4.5. Performance Analysis Applying Robust Tapered Technique

All the tapered beamformers discussed above are conventional types. Those beamformers can reduce SLL levels but cannot attenuate directional interferences. An optimal beamformer is used to detect and attenuate the directional interferences.

However, the performance of an optimal LCCAA beamformer can be degraded due to look direction error. In this regard, we have to find a technique that provides better performance. So, a robust tapered beamformer based on the hamming-VDL technique is proposed in this paper. Our target is to reduce the SLL to a great extent. From Figure 14, one can observe that a tapered robust LCCAA beamformer can detect and attenuate directional interference which is considered at 30°, and also can reduce SLL in comparison to the conventional beamformer. Here conventional, Optimal, FDL, ODL and VDL techniques are used for showing the output power for the hamming window. The output power for Hamming window are −0.589 dB, −55.474 dB, −56.517 dB, −2.0503 dB, and −1.344 dB for conventional, optimal, FDL, ODL, and VDL techniques respectively.

Figure 15 is a comparison of Conventional, robust tapering (VDL- hamming technique) and robust VDL technique. The steering angle is considered at 0° and the actual signal is considered at 2°. The interference direction is assumed at −25°. The signal frequency is taken as 300 MHz.

From Figure 15, it is clear that there is a sharp attenuation at interference direction for robust and robust tapering technique while conventional technique cannot show any kind of attenuation at interference angle. The robust tapered technique shows the better output power of −103.53 dB than the robust and conventional technique at interference angle −25° which indicates the superiority of interference detection capability of the robust tapered technique. It is also clear that the sidelobe level performance is satisfactory in this proposed technique.

Table 7 shows the SLL reducing using different technique for the proposed LCCAA beamformer.

From Table 7 it is clear that a robust tapered technique shows better performance for the LCCAA beamformer. The robust tapered technique consumes around 69.92 dB less sidelobe compared to the conventional technique.

## 5. Conclusions

In this paper, a beamformer named linearly arranged concentric circular antenna array (LCCAA), has been proposed which has been compared with existing geometrically identical beamformers. The proposed LCCAA beamformer has shown better performance with the conventional technique. However, the conventional technique cannot detect the interference and the optimal technique has been applied to peak the actual signal and detect the interference. The optimal technique has a higher SLL with the presence of look direction error. This proposed beamformer has been further analyzed by applying robust techniques- ODL, FDL, and VDL which have reduced beamwidth but the sidelobe level performance remain poor. In this regard, tapering techniques have been applied to reduce the sidelobe level where the hamming tapering technique has the optimum capability for the reduction of sidelobe. In this research work, the robust tapered (VDL-hamming) technique has been introduced by the combination of robust and tapering techniques. It has been observed that the robust tapering technique provides better performance and reduced around 48.39dB and 69.92 dB more SLL compared to conventional and robust techniques. Moreover, the robust tapered technique provides the least beamwidth of 33.11°. Finally, analyzing the simulation results, we can conclude that the proposed LCCAA beamformer is better than any other circular structured beamformer and provides optimum performance in reducing the beamwidth, sidelobe level, and interference cancellation with the proposed robust tapered (VDL-hamming) technique.

## Figures and Tables

**Figure 1 micromachines-13-01959-f001:**
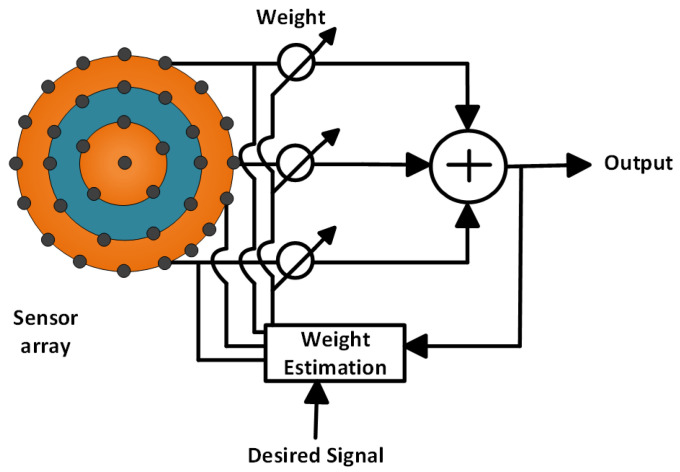
Block diagram of the concentric circular antenna array-based communication system.

**Figure 2 micromachines-13-01959-f002:**
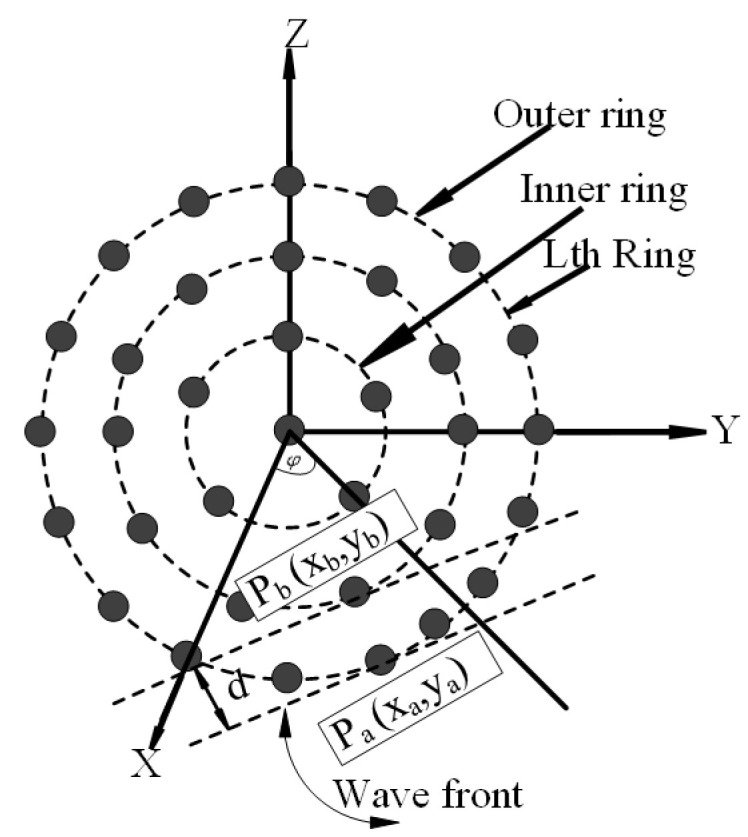
The Geometry of centered element concentric circular antenna array.

**Figure 3 micromachines-13-01959-f003:**
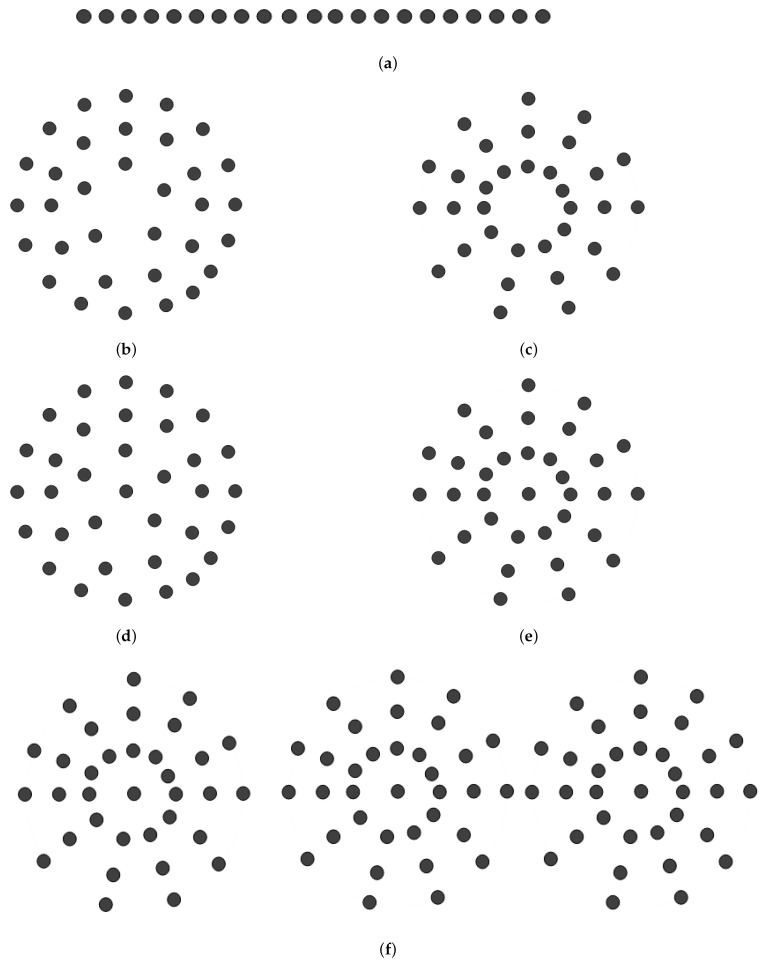
Antenna arrays (**a**) Linear antenna array (21 elements). (**b**) CCAA antenna array (33 elements). (**c**) UCCAA antenna array (33 elements). (**d**) CECCAA antenna array (34 elements). (**e**) UCECCAA antenna array (34 elements). (**f**) ULCCAA antenna array (102 elements).

**Figure 4 micromachines-13-01959-f004:**
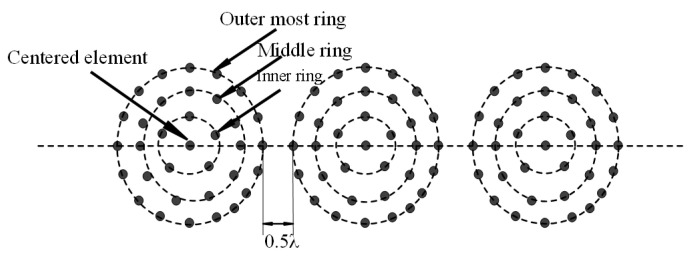
LCCAA antenna array (102 elements).

**Figure 5 micromachines-13-01959-f005:**
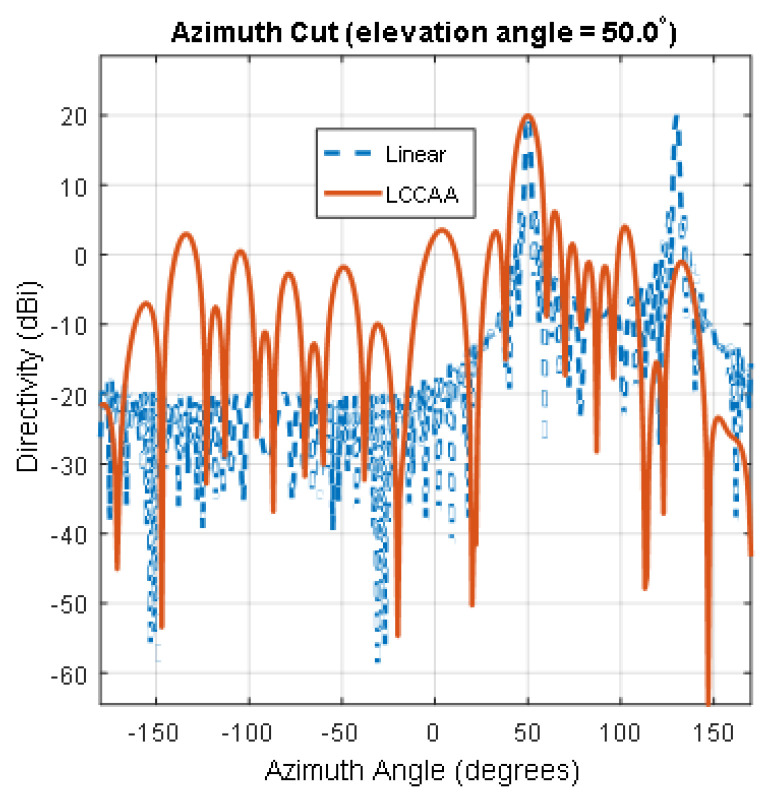
Comparison of directivity between linear beamformer and LCCAA beamformer using equal elements.

**Figure 6 micromachines-13-01959-f006:**
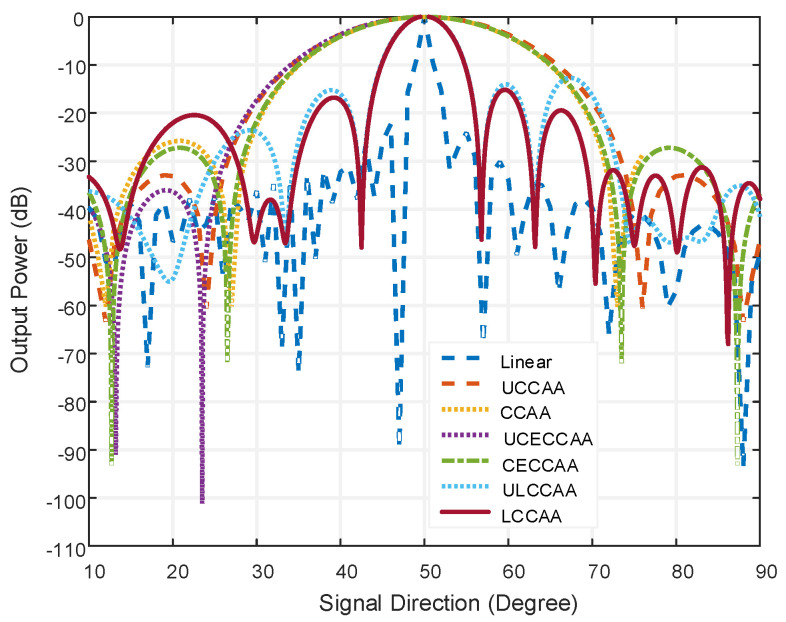
Comparison of all structure having equal elements, applying conventional technique.

**Figure 7 micromachines-13-01959-f007:**
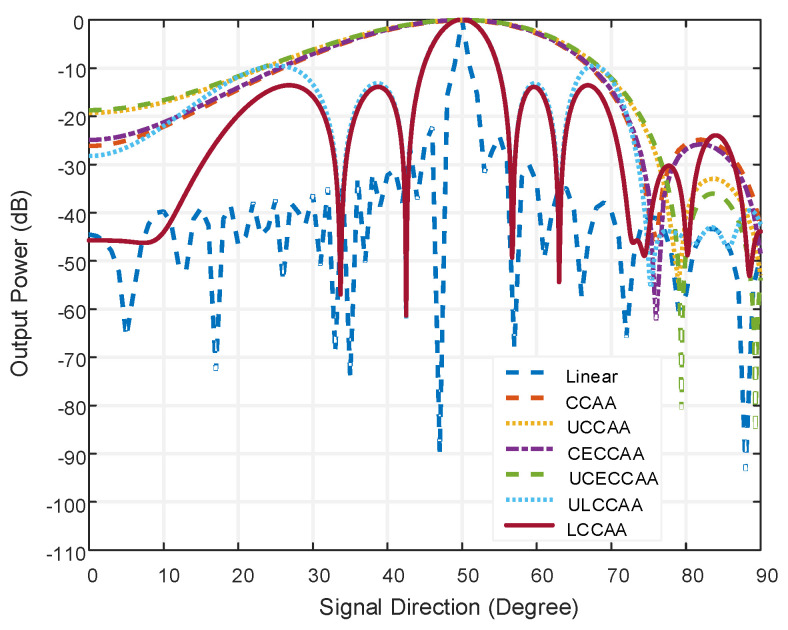
Comparison of all structure having equal element for orthogonal azimuthal plane applying conventional technique.

**Figure 8 micromachines-13-01959-f008:**
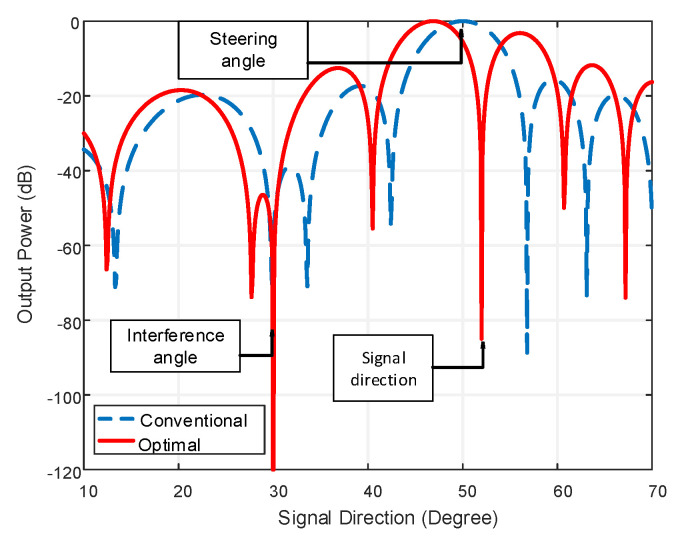
Comparison of Power pattern between conventional and optimal technique with interference and mismatch for the proposed LCCAA beamformer.

**Figure 9 micromachines-13-01959-f009:**
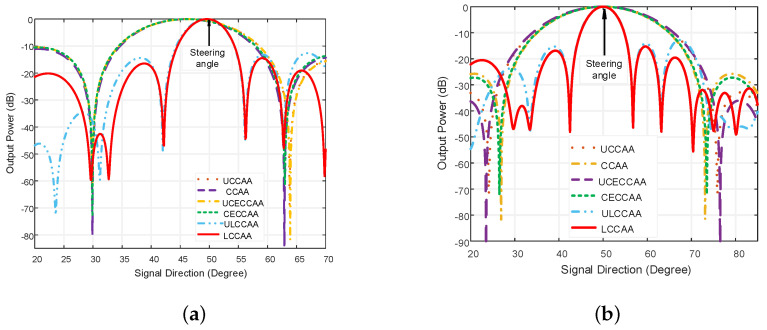
Comparison of all structures for equal element using (**a**) ODL, (**b**) VDL technique.

**Figure 10 micromachines-13-01959-f010:**
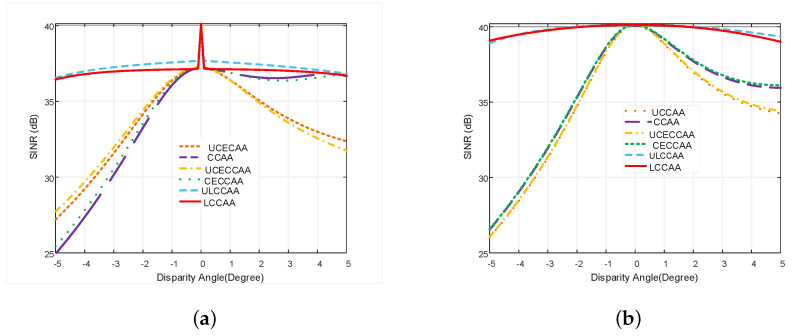
SINR comparison of all structures with equal element based on (**a**) ODL, (**b**) VDL robust technique with respect to disparity angle.

**Figure 11 micromachines-13-01959-f011:**
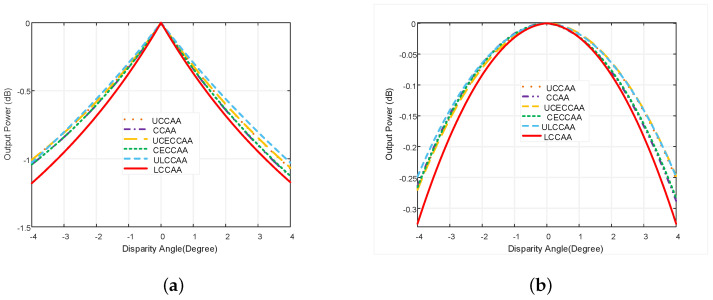
Power pattern comparison of all structures with equal element based on (**a**) ODL and (**b**) VDL robust technique with respect to disparity angle.

**Figure 12 micromachines-13-01959-f012:**
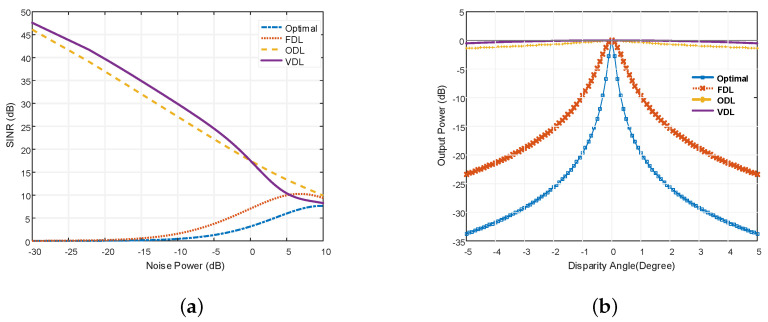
Comparison of (**a**) Output SINR of LCCAA based robust beamformers with the variation of input SNR and noise power (**b**) Power pattern for LCCAA based optimal, FDL, ODL and VDL beamformer.

**Figure 13 micromachines-13-01959-f013:**
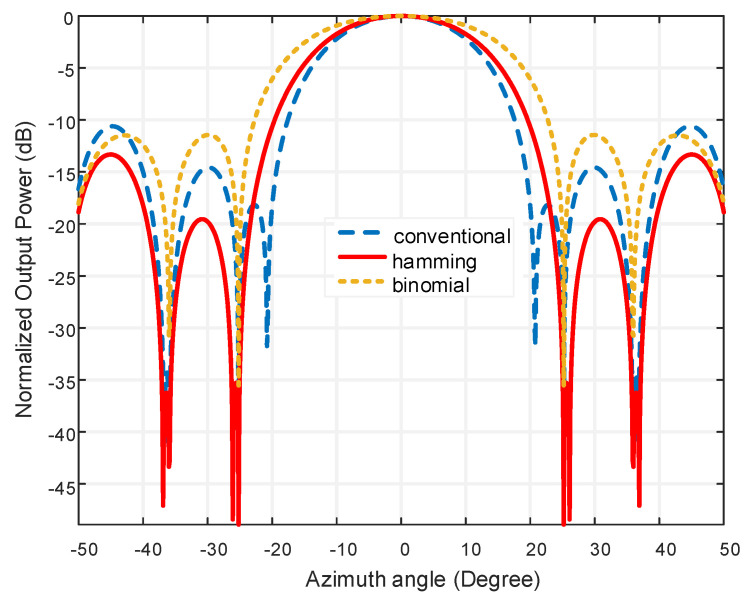
Output power comparison among Hamming, binomial and conventional technique.

**Figure 14 micromachines-13-01959-f014:**
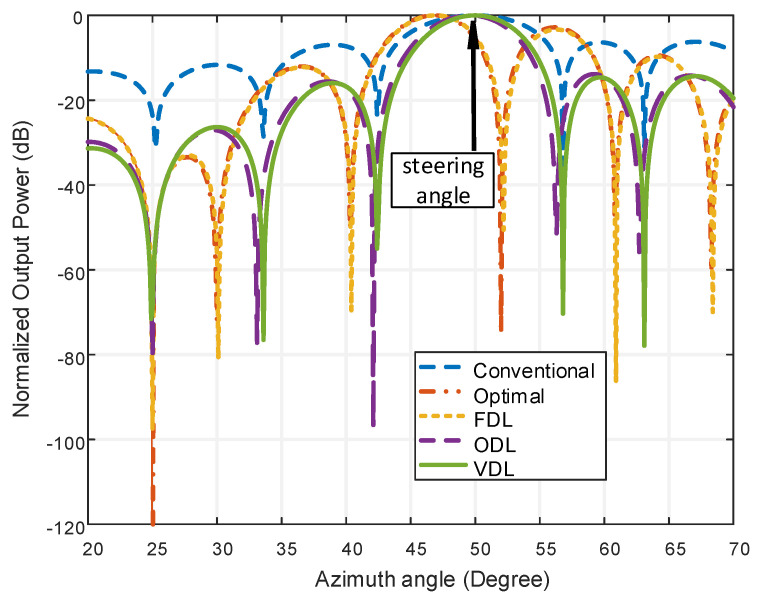
Power pattern of proposed robust tapered LCCAA by using Hamming windows.

**Figure 15 micromachines-13-01959-f015:**
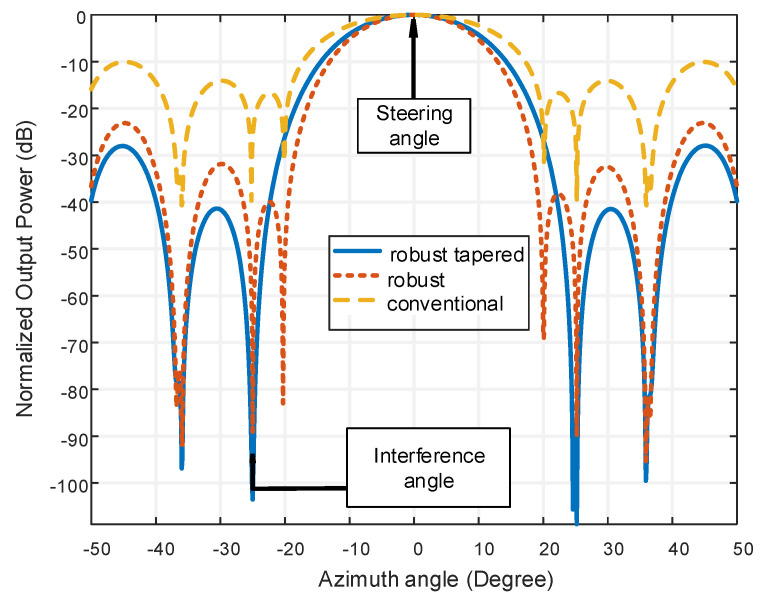
Output power comparison of conventional, robust VDL and robust tapering (VDL-hamming) technique in the LCCAA beamformer.

**Table 1 micromachines-13-01959-t001:** Comparison of beamwidth (BW) and sidelobe level (SLL) among antenna arrays with equal element and identical element spacing by conventional technique.

Beamforming	BW with Normal Plane (Degree)	BW with Azimuth Plane (Degree)	SLL with Normal Plane (dB)	SLL with Azimuth Plane (dB)
UCECCAA [32]	42.62	73.89	−35.75	−36.04
UCCAA [34]	41.67	73.4	−32.90	−32.95
CECCAA [32]	39.35	60.22	−27.78	−25.78
CCAA [33]	38.60	58.98	−25.75	−24.83
Linear [2]	4.64	4.35	−24.08	−24.084
LCCAA [Proposed]	12.50	12.56	−15.17	−13.89

**Table 2 micromachines-13-01959-t002:** Comparison of beamwidth and sidelobe level among antenna arrays with equal elements by robust techniques.

Beamformer	B.W of ODL (Degree)	B.W of VDL (Degree)	SLL of ODL (dB)	SLL of VDL (dB)
UCECCAA [32]	30.07	38.32	−14.44	−35.58
UCCAA [34]	30.02	38.15	−10.38	−35.95
CECCAA [32]	29.33	36.78	−10.49	−26.56
CCAA [33]	29.28	36.63	−10.43	−25.33
ULCCAA	12.46	12.66	−13.164	−14.02
LCCAA [Proposed]	12.34	12.482	−14.472	−15.27

**Table 3 micromachines-13-01959-t003:** Comparison of beamwidth and sidelobe level among antenna arrays with equal elements by robust techniques.

	ODL Robust Technique				VDL Robust Technique			
**Beamformer**	**Without Disparity**	**1° Disparity**	**2° Disparity**	**3° Disparity**	**Without Disparity**	**1° Disparity**	**2° Disparity**	**3° Disparity**
ULCCAA	0	−0.295	−0.56	−0.804	0	−0.018	−0.064	−0.141
CECCAA [32]	0	−0.33	−0.604	−0.84	0	−0.0173	−0.067	0.156
CCAA [33]	0	−0.33	−0.605	−0.84	0	−0.017	-0.066	−0.148
UCCAA [34]	0	−0.315	−0.581	−0.81	0	0.02	0.071	−0.153
UCECCAA [32]	0	−0.32	−0.581	−0.81	0	−0.021	−0.072	−0.156
LCCAA [Proposed]	0	−0.373	−0.69	−0.951	0	−0.023	−0.08	−0.183

**Table 4 micromachines-13-01959-t004:** Output power of the proposed beamformer using Optimal, FDL, ODL and VDL technique at different diaparity angle.

Robust Techniques	Without Disparity	1° Disparity	2° Disparity	3° Disparity
Optimal	0	−19.65	−25.66	−29.21
FDL	0	−9.68	−15.36	−18.85
ODL	0	−0.38	−0.69	−0.96
VDL	0	−0.02	−0.09	−0.19

**Table 5 micromachines-13-01959-t005:** The sidelobe level comparison of different tapering techniques for LCCAA beamformer with the increase of ring number.

Ring No.	Uniform	Binomial	Blackman	Chebyshev	Taylor	Hamming	Hanning	Triangular
1	−6.414	−8.615	−8.393	−12.568	−13.126	−7.344	−7.461	−10.320
2	−16.396	−10.859	−23.937	−21.126	−23.449	−28.472	−33.596	−25.727
3	−18.182	−11.441	−18.183	−22.063	−23.447	−35.279	−21.074	−22.500
4	−18.935	−10.907	−17.291	−22.852	−23.528	−27.132	−23.414	−21.507
5	−19.347	−10.399	−16.436	−23.482	−23.580	−24.342	−21.335	−20.934
6	−19.606	−9.978	−15.863	−24.000	−23.613	−22.819	−20.120	−20.546
7	−19.795	−9.635	−15.449	−24.433	−23.643	−21.836	−19.304	−20.267
8	−19.910	−9.353	−15.131	−24.810	−23.653	−21.150	−18.716	−20.043
9	−20.0282	−9.121	−14.883	−25.117	−23.679	−20.644	−18.264	−19.865
10	−20.1	−8.925	−14.685	−25.412	−23.695	−20.244	−17.908	−19.725

**Table 6 micromachines-13-01959-t006:** The beamwidth comparison of different tapering windows for proposed LCCAA beamformers with increase of ring number.

Ring No.	Uniform	Binomial	Blackman	Chebyshev	Taylor	Hanning	Hamming	Triangular
1	60.692	70.037	63.440	66.908	67.204	62.253	62.090	65.364
2	32.898	43.264	37.570	37.896	37.462	35.746	35.378	37.060
3	22.830	31.769	27.211	26.263	26.210	25.662	25.242	25.995
4	17.516	25.340	23.298	20.038	19.934	20.678	20.960	20.340
5	14.226	20.938	17.730	16.176	16.174	16.660	16.240	16.620
6	11.980	17.916	15.120	13.540	13.612	14.220	13.816	14.084
7	10.348	15.662	13.194	11.658	11.752	12.414	12.026	12.214
8	9.108	13.917	11.703	10.226	10.341	11.021	10.648	10.783
9	8.135	12.522	10.516	9.105	9.233	9.910	9.557	9.653
10	7.348	11.384	9.547	8.206	8.338	9.004	8.668	8.738

**Table 7 micromachines-13-01959-t007:** The reducing of sidelobe level of proposed LCCAA beamformer for all technique.

Beamforming Technique	Sidelobe Level (dB)
Conventional	−16.70
Robust (VDL)	−38.23
Robust Tapered (proposed)	−86.62

## Data Availability

Not applicable.

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
