# Peer review of "Performance Analysis of Linearly Arranged Concentric Circular Antenna Array with Low Sidelobe Level and Beamwidth Using Robust Tapering Technique"

_micromachines, 2022, doi:10.3390/mi13111959_

Round 1

Reviewer 1 Report

The paper presents a very solid design of LCCAA array having low SLL and narrow beam width. In general, there is not much novelty in the design.

1. First of all, I would suggest improving the English writing especially the grammar.
2. The paper is too long, it includes the very basic definition of the antenna and arrays theory, which is not necessary. For example, all the equations on page 9 and 10 can be find in antenna text books.
3. Why grating lobes are higher in LCCAA than the ones in ULCCAA.
4. Please describe the conclusion from each table in a paragraph.
5. The size of the graphs changes non-uniform. Fig 9 is too big, while Fig. 13 is small. Remove extra gaps between Fig. 10 and 11, any other places.
6. From the figures shown, the value given in Table 7 corresponds to a null, but it is called SLL!

Author Response

1.First of all, I would suggest improving the English writing especially the grammar.
Response: It has been improved in current manuscript.
2. The paper is too long, it includes the very basic definition of the antenna and arrays theory, which is not necessary. For example, all the equations on page 9 and 10 can be find in antenna text books.
Response: Page 9 and 10, equations has been removed.
3. Why grating lobes are higher in LCCAA than the ones in ULCCAA.
Response: This issue has been taken care of in the current manuscript by including the following sentence. When compared to an LCCAA beamformer, the ULCCAA offers superior sidelobe level performance. In terms of structure and performance, the ULCCAA beamformer is close to the LCCAA beamformer. ULCCAA beamformer can therefore works well as an antenna array. For a conclusion, more investigation is required.
4. Please describe the conclusion from each table in a paragraph.
Response: it has been done very carefully.
5. The size of the graphs changes non-uniform. Fig 9 is too big, while Fig. 13 is small. Remove extra gaps between Fig. 10 and 11, any other places.
Response: Now the graph size is similar in current manuscript.
6. From the figures shown, the value given in Table 7 corresponds to a null, but it is called SLL!
Response: From the figure 15 the value of SLL has ben taken. Table 7 mainly clarifies  the superiority applying proposed robust tapered beamforming technique.

Reviewer 2 Report

The paper must be thoroughly revised in terms of writing. The English syntax is bad at many parts of the manuscript. Also, the introduction and length of the paper must be reduced.

Too few examples are given to demonstrate the superiority of the proposed scheme. To prove the superiority some averages should be given after performing a big number of tests and then presenting the average SLL, main lobe divergence and nulls divergence compared to the other methods. 

Author Response

The paper must be thoroughly revised in terms of writing. The English syntax is bad at many parts of the manuscript. Also, the introduction and length of the paper must be reduced. Too few examples are given to demonstrate the superiority of the proposed scheme. To prove the superiority some averages should be given after performing a big number of tests and then presenting the average SLL, main lobe divergence and nulls divergence compared to the other methods. 
Response; We thoroughly revised the current manuscript. English grammar has been improved. We carefully address the issues of this report. 

Reviewer 3 Report

1. is this antenna fabricated and tested or not? 

2. what is the lob of antenna

3. What is the simulation platform...like HFSS or any other? I could not find anything related.

4. there is only a few simulations that don't reflect the antenna performance

5. What is Eq 11.. a lot of equations are not explained

Author Response

  1.  is this antenna fabricated and tested or not?

Response: The antenna is not fabricated. The result is based on simulation result.

  1. what is the lob of antenna

Response: We tried to find out  best techniques to reduce the SLL as much as possible. 

  1. What is the simulation platform...like HFSS or any other? I could not find anything related.

response: MATLAB is the simulation platform.

  1. there is only a few simulations that don't reflect the antenna performance

response: this research article  is all about array signal processing. We tried to minimize the SLL as much as possible.

  1. What is Eq 11.. a lot of equations are not explained

Response; It has been taken care of.

Round 2

Reviewer 1 Report

The paper presents a very solid design of LCCAA array having low SLL and narrow beam width. In general, there is not much novelty in the design. 1. First of all, I would suggest improving the English writing especially the grammar. 2. The paper is too long, it includes the very basic definition of the antenna and arrays theory, which is not necessary. For example, all the equations on page 9 and 10 can be find in antenna text books. 3. Why grating lobes are higher in LCCAA than the ones in ULCCAA. 4. Please describe the conclusion from each table in a paragraph. 5. The size of the graphs changes non-uniform. Fig 9 is too big, while Fig. 13 is small. Remove extra gaps between Fig. 10 and 11, any other places. 6. From the figures shown, the value given in Table 7 corresponds to a null, but it is called SLL!

Author Response

First of all, I’d like to pay my gratitude to the reviewer for spending his valuable time to review our manuscript. In this response letter, We will discuss how we address each concern raised by the reviewer.

Reviewer 1:

Concern 1:  First of all, I would suggest improving the English writing especially the grammar.

Response: We have modified the English writing in our current manuscript. We have carefully checked the English writing via paid Grammarly software.

Concern 2. The paper is too long, it includes the very basic definition of the antenna and arrays theory, which is not necessary. For example, all the equations on page 9 and 10 can be find in antenna text books.

Response: Yes, we have eradicated almost all the basic antenna array theory in our current manuscript. All equations from pages 9 and 10 are removed in the present manuscript.

Concern 3. Why grating lobes are higher in LCCAA than the ones in ULCCAA.

Response: The following sentences have been added to the current manuscript to address this issue.

When compared to an LCCAA beamformer, the ULCCAA offers superior sidelobe level performance. In terms of structure and performance, the ULCCAA beamformer is close to the LCCAA beamformer. ULCCAA beamformer can therefore work well as an antenna array. For a conclusion, more investigation is required”.

Concern  4. Please describe the conclusion from each table in a paragraph.

Response: Yes, the findings from each table have been discussed in a paragraph right after the table.

Concern  5. The size of the graphs changes non-uniform. Fig 9 is too big, while Fig. 13 is small. Remove extra gaps between Fig. 10 and 11, and any other places. 6.

Response:  We made every attempt to maintain the size of each figure consistent throughout this article and to eliminate any unnecessary gaps.

Concern  6: From the figures shown, the value given in Table 7 corresponds to a null, but it is called SLL!

Response: From figure 15 the value of Side Lobe Level (SLL) has been taken. Table 7 mainly clarifies the superiority of the proposed robust tapered beamforming technique.

Reviewer 2 Report

1. There is no point to point response letter regarding the reviewers' comments.

2. The issues specified by the reviewer are not addressed in the revised version.

Author Response

Review Response Letter

First of all, I’d like to pay my gratitude to the reviewer for spending his valuable time reviewing our manuscript. In this response letter, We will discuss how we address each concern raised by the reviewer.

Concern 1:  The paper must be thoroughly revised in terms of writing. The English syntax is bad at many parts of the manuscript. Also, the introduction and length of the paper must be reduced.

Response: The English writing in our present manuscript has been changed. Using the premium Grammarly software, we carefully reviewed the English writing. Additionally, we have reduced the article by eliminating all the basic equations, unnecessary gaps, and introductory lines.

Concern 2. Too few examples are given to demonstrate the superiority of the proposed scheme. To prove the superiority some averages should be given after performing a big number of tests and then presenting the average SLL, main lobe divergence, and nulls divergence compared to the other methods. 

Response: This manuscript is actually based on the simulation result in the MATLAB platform. We didn’t fabricate anything and take the data from the practical experiments. In fact, we don’t have such facilities right now to perform it practically. Additionally, our manuscript is already quite long; adding the average SLL, main lobe divergence, and nulls will increase the size. In our manuscript, we have shown the SLL data to prove the superiority of our proposed beamforming techniques.

Reviewer 3 Report

Please review the paper for English grammar and typos

Author Response

Review Response Letter

First of all, I’d like to pay my gratitude to the reviewer for spending his valuable time reviewing our manuscript. In this response letter, We will discuss how we address each concern raised by the reviewer.

Concern 1:  Please review the paper for English grammar and typos.

Response: The English writing in our present manuscript has been improved. Using the premium Grammarly software, we carefully reviewed the English writings. Additionally, we have reduced the article size by eliminating all the basic equations, unnecessary gaps, and introductory lines.

Round 3

Reviewer 2 Report

All queries are addressed in this review round.